

# Back to the formula – LHC edition

Anja Butter[1], Tilman Plehn[1], Nathalie Soybelman[1]* and Johann Brehmer[2]

**1** Institut für Theoretische Physik, Universität Heidelberg, Germany
**2** Center for Data Science, New York University, New York, United States

* nathalie@soybelman.de

## Abstract

While neural networks offer an attractive way to numerically encode functions, actual formulas remain the language of theoretical particle physics. We use symbolic regression trained on matrix-element information to extract, for instance, optimal LHC observables. This way we invert the usual simulation paradigm and extract easily interpretable formulas from complex simulated data. We introduce the method using the effect of a dimension-6 coefficient on associated ZH production. We then validate it for the known case of CP-violation in weak-boson-fusion Higgs production, including detector effects.

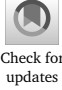

# 1 Introduction

The defining feature of modern LHC physics is the combination of fundamental physics questions, precision simulations based on first-principle quantum field theory, and state-of-the-art statistics and analyses. In the ideal LHC world we will use likelihood-free or simulation-based inference [1] to compare simulated data sets with recorded data sets and extract fundamental physics parameters using likelihood methods. Machine learning has the potential to transform many parts of this analysis chain, from enabling faster and more precise simulations to triggering events, providing stable and economic analysis objects, to the actual inference. On the other hand, a fundamental physics description in terms of perturbative quantum field theory often allows us to write down compact and instructive mathematical expressions for scattering amplitudes or observables. This advantage is lost when we turn to numerical methods like neural networks.

The way to combine the power of machine learning with the advantage of mathematical intuition is symbolic regression. In analogy to training a neural network we can use this method to learn a general, analytic function over phase space from a data set. While the standard methodology in particle physics is to start from human-readable formulas and build numerical simulations on them, symbolic regression allows us to invert this method and extract human-readable formulas from simulated data sets. If the performance of this function is comparable to the numerically trained network, such an analytic expression represents the best of both worlds and can trigger fundamental considerations explaining the approximate analytic formula. In this paper we approximate numerically defined optimal observables, or scores, for simple LHC processes with closed formulas and show how those compare to known fundamental properties and expressions.

One of the most pressing physics questions for the LHC is the properties of the Higgs boson, the currently only fundamental scalar particle [2]. The theory framework for Higgs analyses is the Standard Model Effective Field Theory (SMEFT) [3], which combines rate information and kinematic distributions in global analyses [4–10]. Given a set of Wilson coefficients describing physics beyond the Standard Model, the straightforward question is how we can best measure a specific model parameter in a specific LHC process. In the usual LHC analysis framework of theory-inspired observables this leads to the problem of finding the *optimal observable* to measure a given parameter in a given process [11–14]. At the detector level, optimal observables or *scores* [15] can be encoded in form of neural networks [16–18], automated in the MADMINER library [19]. They have proven useful in different applications to LHC Higgs physics [20–23].

In this paper we use symbolic regression [24–27] to construct optimal observables for LHC processes as human-interpretable formulas. We rely on MADMINER [19] to extract matrix-element information from simulated events and on PYSR [28] to approximate the score as a closed-form symbolic expression. We then show how the so-defined observables compare to established fundamental properties and expressions. Unlike the traditional parton-level method, our approach allows us to incorporate backgrounds, jet radiation, and detector effects. Unlike the neural approach, its output is a human-readable expression such as $p_{T,1} p_{T,2} \cos(\Delta\phi_{jj})$.

After introducing all relevant concepts and tools in Sec. 2, we will illustrate how symbolic regression can learn the optimal observable for the Wilson coefficient $f_B$ in $ZH$ production in Sec. 3. For this simple on-shell scattering process, we discuss possible functional forms and a suitable modification of the standard PYSR algorithm. In Sec. 4, we will apply symbolic regression to determine the optimal observable for the $CP$-violating Wilson coefficient $f_{W\widetilde{W}}$ in weak-boson-fusion (WBF) Higgs production. In this case we know the analytic form for small Wilson coefficients at parton level [21,29,30], it has been shown to work in actual analyses [31–34], so we will benchmark our symbolic regression approach and study the case of

larger Wilson coefficients and detector effects. Finally, we compare the expected performance of our approximate formulas to the complete numerical MADMINER output.

## 2 Basics

### 2.1 Optimal observables or score

Historically, LHC analyses identify phase space regions with a large signal-to-background ratio and focus on them by applying cuts on kinematic observables. A measurement is then based on counting events and comparing this rate to the background-only and the signal-plus-background predictions. Such an analysis has the fundamental disadvantage that there will always be kinematic observables and phase space regions which do not contribute to our task. One way to improve these analyses is to change the way we organize events. Instead of a simple kinematic observable, we can define histograms in terms of any variable we want, and we can systematically construct optimal test statistics for a given task.

The central object for constructing an optimal observable or score is the likelihood function for a single event at the LHC,

$$p(x|\theta) = \frac{1}{\sigma_{\text{tot}}(\theta)} \frac{\text{d}^d \sigma(x|\theta)}{\text{d}x^d}. \tag{1}$$

The symbol $x$ stands for all of the information we observe for an event, for instance as a vector in terms of a basis of observables, including particle IDs of reconstructed particles. $\theta$ is the vector of theory parameters of interest, $\text{d}^d \sigma(x|\theta)/\text{d}x^d$ is the fully differential cross section, and $\sigma_{\text{tot}}$ is the total cross section. If we are interested in parameter values $\theta$ close to a reference point $\theta_0$, we can taylor the log likelihood around $\theta_0$,

$$\log \frac{p(x|\theta)}{p(x|\theta_0)} = (\theta - \theta_0) \cdot \underbrace{\nabla_\theta \log p(x|\theta)\Big|_{\theta_0}}_{t(x|\theta_0)} + \cdots \tag{2}$$

The first-order term in this expansion is known as the score in the field of statistics [15]. If the second-order term is negligible, we can solve this equation and find

$$p(x|\theta) \approx e^{t(x|\theta_0) \cdot (\theta - \theta_0)} p(x|\theta_0). \tag{3}$$

This likelihood function has the property that $t(x|\theta_0)$ are its sufficient statistics; measuring this score contains all of the information on the parameters $\theta$ as the full event record $x$. In the vicinity $\theta \sim \theta_0$ we can then define an optimal observable for each model parameter $\theta_i$ as [12],

$$\mathcal{O}_i^{\text{opt}}(x) \equiv t(x|\theta_0) = \frac{\partial \log p(x|\theta)}{\partial \theta_i}\Big|_{\theta_0}. \tag{4}$$

From Eq.(2) we also see that it is optimal in the sense that it approximates the log-likelihood ratio as the optimal discriminator. Using the same simplifying assumptions, it is possible to show that the score or optimal observable is not only linked to the Neyman-Pearson lemma [35], but also saturates the Cramér-Rao bound [36, 37], for a particle physics-related discussion see *e.g.* Refs. [2, 20, 38] and indeed includes all available information on a continuous model parameter.

For many LHC applications, including measuring SMEFT Wilson coefficients, a natural reference point is the Standard Model with $\theta_0 = 0$. At parton level and assuming all particle

properties can be observed, the likelihood is proportional to the transition amplitude and we find

$$p(x|\theta) \sim |\mathcal{M}|_0^2 + \sum_n \theta_n |\mathcal{M}|_{\text{int},n}^2 + \mathcal{O}(\theta^2) \qquad \Rightarrow \qquad \mathscr{O}_i^{\text{opt}}(x) \equiv t(x|\theta_0) \propto \frac{|\mathcal{M}|_{\text{int},i}^2}{|\mathcal{M}|_0^2}, \qquad (5)$$

where we have omitted additive and multiplicative constants.

Computing the score $t(x|\theta_0)$ beyond parton level is not straightforward, because the likelihood function $p(x|\theta)$ is, in general, intractable. However, it is linked to the scattering matrix elements in that the single-event likelihood of Eq.(1) can be written as [16–18]

$$p(x|\theta) \propto \int dz \, p(x|z) |\mathcal{M}(z|\theta)|^2, \qquad (6)$$

where we integrate over the full parton-level information $z$, $|\mathcal{M}(z|\theta)|^2$ is the squared matrix element evaluated for parameters $\theta$, and $p(x|z)$ relates the full parton-level information $z$ to the observables $x$, including parton shower and detector effects.

For a simulated event, we know the complete parton-level information $z$ and can compute the *joint score*

$$t(x,z|\theta) = \frac{\nabla_\theta |\mathcal{M}(z|\theta)|^2}{|\mathcal{M}(z|\theta)|^2} - \frac{\nabla_\theta \sigma_{\text{tot}}(\theta)}{\sigma_{\text{tot}}(\theta)}. \qquad (7)$$

This joint score is not useful, since it depends on unobserved parameters as part of $z$. However, it turns out that the score $t(x|\theta)$ can be linked to the joint score $t(x,z|\theta)$ as the minimum of the mean-squared-error functional:

$$t(x|\theta) = \arg\min_{g(x)} \mathbb{E}_{x,z \sim p(x,z|\theta)} |g(x) - t(x,z|\theta)|^2. \qquad (8)$$

In practice, we can perform this minimization by choosing an expressive variational family for $g(x)$ and fitting its parameters to simulated data.

The first instantiation of this idea is the SALLY method [16–18], which uses a neural network as fitting function $g(x)$ and learns its parameters through stochastic gradient descent. In this work we propose an alternative approach: for $g(x)$, we use a set of symbolic expressions, closed-form formulas that combine elementary elements and simple functions in a human-readable way. For this purpose we minimize the loss functional in Eq. (8) with a genetic algorithm.

## 2.2 MadMiner

To generate LHC events for finite Wilson coefficients we use the reweighting option in MADGRAPH5, combined with the known, quadratic dependence of the production cross section on the Wilson coefficient. This gives us event weights for different values of the Wilson coefficient, which are then extracted by MADMINER 0.5 [19] and used for the calculation of the joint score via a morphing technique [17].

The joint score is essentially extracted from the 4-momenta of the outgoing particles at parton level. Taking the joint score, the neural net SALLY can be used to regress the score on the real kinematic observables after shower and detector. The goal of this paper is to replace the neural network by an explicit analytic formula obtained through the symbolic regression tool PYSR.

We use 500k events from the MADGRAPH5 [39], PYTHIA8 [40], and DELPHES [41] simulation chain with the default CMS card. With MADMINER we extract matrix-element information from our Monte-Carlo simulations and calculate the expected limits on the Wilson coefficients.

Additionally we use the implemented SALLY algorithm, trained on the same events, as a baseline for comparison with symbolic regression results. For the network training we rely on the AMSGRAD optimizer [42].

## 2.3   Symbolic regression

For our symbolic regression we rely on PYSR [28]. It uses genetic programming to find a symbolic expression for a numerically defined function in terms of pre-defined variables. The population consists of symbolic expressions, visualized as a tree and consisting of nodes with an operator function or an operand. We use the operators for addition, subtraction, multiplication, squaring, cubing and if needed division. The tree population evolves when new individuals are created and old ones are discarded. To breed the next generation, several mutation operators can be applied, for instance exchanging, adding or deleting nodes of the parent tree. The hyperparameter `populations = 30` defines the number of populations and is per default set to the number of processors used (`procs`). The number of individuals per populations is given by `npop = 1000`.

As the figure of merit for the PYSR algorithm we take the mean squared error between the data points $t_i(x,z|\theta)$ and the functional description $g_i$,

$$\text{MSE} = \frac{1}{n}\sum_{i=1}^{n}(g_i(x) - t_i(x,z|\theta))^2 \,, \tag{9}$$

and balance it with the function's complexity, defined as

$$\text{complexity} = \#\text{nodes}. \tag{10}$$

For the PYSR score value, not to be confused with the statistics version of the optimal observable defined in Eq.(2), the parameter `parsimony` defined through

$$\text{score} = \frac{\text{MSE}}{\text{baseline}} + \text{parsimony} \cdot \text{complexity}, \tag{11}$$

balances the two conditions. The normalization factor `baseline` is the MSE between the data and the constant unit function. The hyperparameter `maxsize` restricts the complexity to a maximum value. We adjust this value depending on the difficulty of the regression task taking 50 as a starting point and increase (decrease) it if the required complexity is larger (smaller). Additionally we can restrict the complexity of specific operators to obtain a more readable result. We set the maximal complexity of square to 5 and cube to 3. Note that in some instances we choose to not extract the score, but the score scaled by a constant, to improve the numerics with an order-one function.

Simulated annealing [43] allows us to search for a global optimum in a high-dimensional space while preventing the algorithm from being stuck in a local optimum. A mutation is accepted with the probability

$$p = \exp\left(-\frac{\text{score}_{\text{new}} - \text{score}_{\text{old}}}{\text{alpha} \cdot T}\right). \tag{12}$$

The parameter $T$ is referred to as temperature. It linearly decreases with each cycle or generation, starting with 1 in the first cycle and 0 in the last. The hyperparameter `ncyclesperiterations = 200` sets the amount of cycles. We choose `alpha = 1`. If the new function describes the data better than the reference tree, $\text{score}_{\text{new}} \ll \text{score}_{\text{old}}$, the exponent has a positive sign and the new function is accepted. If the new sore is larger than

the old score, the acceptance of the new function is given by $p$ and hence exponentially suppressed. We use this default PYSR form for our simple example and discuss a better-suited form for our application in Sec. 3.

The hyperparameter `niterations = 300` defines the number of iterations of a full simulated annealing process. After each iteration the best formulas are compared to the hall of fame (HoF). For each complexity the best equation is chosen and saved in the output file. An equation of higher complexity is only added if its MSE is smaller than for previous formulas. Equations from different populations or the hall of fame can migrate to other populations. This process is affected by the parameters `fractionReplaced = 0.5` and `fractionReplacedHof = 0.2`.

## 3 ZH production

To illustrate our symbolic regression task we choose the LHC production process

$$pp \to ZH \,, \tag{13}$$

without decays and modified by a single dimension-6 operator,

$$\mathcal{L} = \mathcal{L}_{\mathrm{SM}} + \frac{f_B}{\Lambda^2} \mathcal{O}_B \,, \qquad \text{with} \qquad \mathcal{O}_B = \frac{ig'}{2}(D^\mu \phi)^\dagger D^\nu \phi B_{\mu\nu} \,. \tag{14}$$

This operator is know to modify the boosted regime of $ZH$ production [22, 44–46]. For our numerical results we quote $f_B$-values for $\Lambda = 1$ TeV.

We generate parton level events with MADGRAPH5 with the EWDIM6 model file [47]. Considering $ZH$ production at parton level and without decays, the number of degrees of freedom is given by two on-shell 3-momenta minus transverse momentum conservation. Of these four degrees of freedom the azimuthal angle is a symmetry, so we expect three observables to describe the effects of $f_B$ over phase space. In Fig. 1 we show distributions for the candidate observables

$$p_{T,Z} = p_{T,H} \,, \qquad \text{and} \qquad \eta_\pm = \eta_Z \pm \eta_H \,, \tag{15}$$

for $f_B = 0, 2, 10$, where the largest value is experimentally ruled out and only chosen for illustration purposes. At first sight the Wilson coefficient seems to affect $p_T$ and $\eta_+$, while $\eta_-$ looks insensitive. However, this is an artifact of looking at 1-dimensional histograms. In Fig. 2 we show the correlation between $\eta_+$ and $\eta_-$ in slices of $p_T$. In the right column, the ratio indicates that for given $p_T$ there is no variation in $\eta_+$, except for a smaller global range, which reflects a general suppression of events with, both, large $p_T$ and $p_z$. On the other hand, there is a small residual dependence on $\eta_-$, in that highly boosted events are more central.

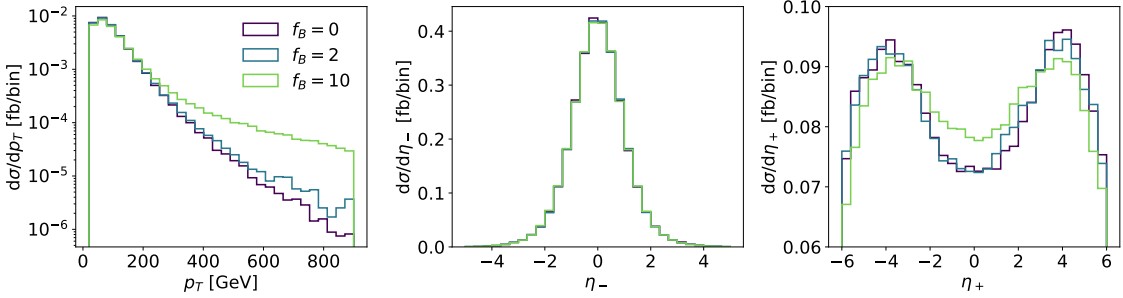

Figure 1: Kinematic distributions for $ZH$ production at parton level with different Wilson coefficients $f_B$. We define $\eta_\pm = \eta_Z \pm \eta_H$.

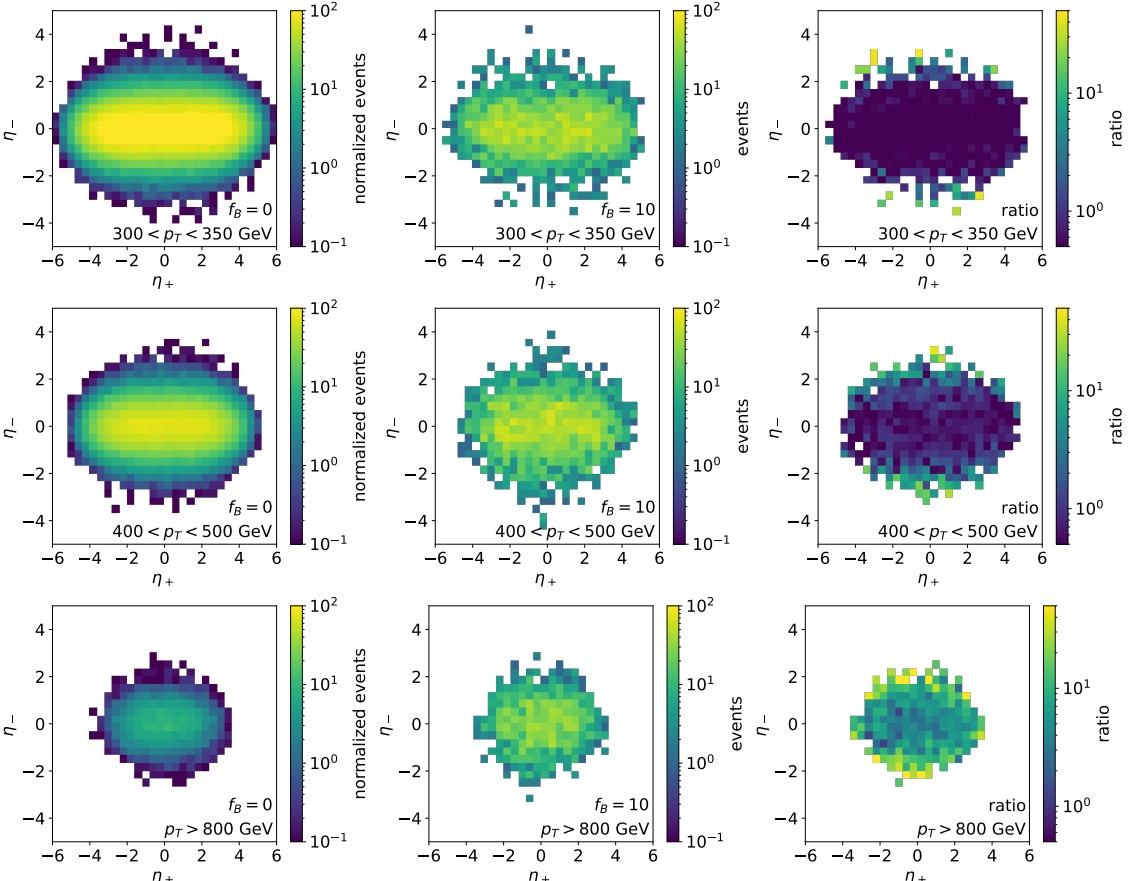

Figure 2: Kinematic $\eta_-$ vs $\eta_+$ correlations for $ZH$ production with $f_B = 0, 10$. We show $p_T$-slices in the boosted regime.

## 3.1 Score for $f_B$

The advantage of our simple $ZH$ example process is that we can analytically compute the score to leading order. We start with the joint score in the presence of unphysical parameters $z$ as given in Eq.(7). The differential cross section for $ZH$ production is

$$\mathrm{d}\sigma(z|\theta) = \frac{(2\pi)^4 f_1(x_1) f_2(x_2)}{8 x_1 x_2 s} |\mathcal{M}|^2 (z|\theta) \, d\Phi(x) , \tag{16}$$

with the momentum fractions $x_i$ of the partons, the squared center-of-mass energy $s$, and the parton densities $f_i(x_i)$. If the matrix element is quadratic in the Wilson coefficient we can write it as

$$|\mathcal{M}(\theta)|^2 \sim p_0 + a\theta + b\theta^2 , \tag{17}$$

and find for the first term in Eq.(7)

$$\frac{\nabla_\theta |\mathcal{M}(\theta)|^2}{|\mathcal{M}(\theta)|^2} = \frac{a + 2b\theta}{p_0 + a\theta + b\theta^2} . \tag{18}$$

We consider two limits for this expression in Tab. 1. For small Wilson coefficients we only keep the leading term in $\theta$ and find that the score decreases as long as $2b < a^2/p_0$. Evaluated around the Standard Model, the contribution to the score is constant, specifically $a/p_0$. For large new physics contributions we neglect the constant and linear terms. In that case the score decreases like $2/\theta$ for increasing $\theta$.

Table 1: Limits for the first term of the joint score in Eq.(7).

| | $\theta \ll 1$ | $\theta \gtrsim 1$ |
|---|---|---|
| approximation | leading term $\dfrac{a}{p_0} + \dfrac{1}{p_0}\left(2b - \dfrac{a^2}{p_0}\right)\theta$ | quadratic term $\dfrac{2}{\theta}$ |
| scaling | mostly constant | decreasing with $\theta$ |

The situation is more complicated for the second term, because the total cross section requires a phase space integration and the prefactors in Eq.(16) do not cancel

$$\frac{\nabla_\theta \sigma_{\text{tot}}(\theta)}{\sigma_{\text{tot}}(\theta)} = \frac{\int d\Phi(x)\, f_1(x_1,Q^2)/x_1\, f_2(x_2,Q^2)/x_2\, (a+2b\theta)}{\int d\Phi(x)\, f_1(x_1,Q^2)/x_1\, f_2(x_2,Q^2)/x_2\, (p_0+a\theta+b\theta^2)}. \tag{19}$$

This contribution is essentially a constant in $\theta$, but it is different for different quark flavors. To simplify our problem we will start by only looking at one quark type in the initial state, allowing us to neglect this score contribution.

For a single quark flavor and only considering the $Z$-contribution,

$$u\bar{u} \to Z^* \to ZH, \tag{20}$$

the partonic squared matrix element has the compact form.

$$|\mathcal{M}|^2 = \frac{2g^2(V^2+A^2)}{c_w^2(s-m_Z^2)^2} x_1 x_2 s \left(2m_Z^2 + p_T^2\right) \left[\frac{m_Z}{v} + \frac{f_B}{\Lambda^2}\frac{g'^2 v}{8m_Z}\left(m_H^2 + 2p_H p_Z\right)\right]^2. \tag{21}$$

Around the Standard Model the linear score contribution of Eq.(18) reads

$$t(x|f_B = 0) \approx \frac{a}{p_0} = \frac{g'^2 v^2}{4m_Z^2}\left(m_H^2 + 2p_H p_Z\right). \tag{22}$$

In Fig. 3 we show the kinematic dependence of the score from our numerical evaluation. In the left panel we see that the $p_T$-dependence of the score is mild for small Wilson coefficients and small $p_T$. For larger $p_T$ we also see the quadratic scaling from the formula. Towards

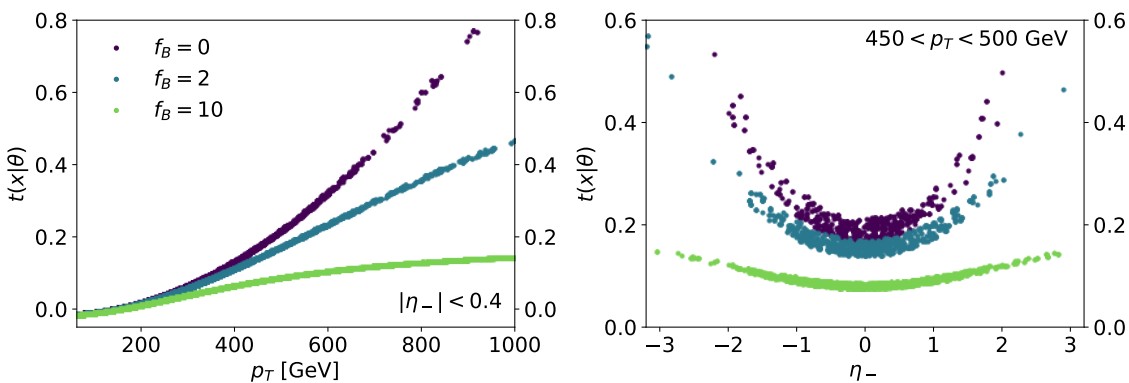

Figure 3: Kinematic distributions, $p_T$ and $\eta_-$, for different values of $f_B$. We only include the $Z$-contribution and one initial parton flavor.

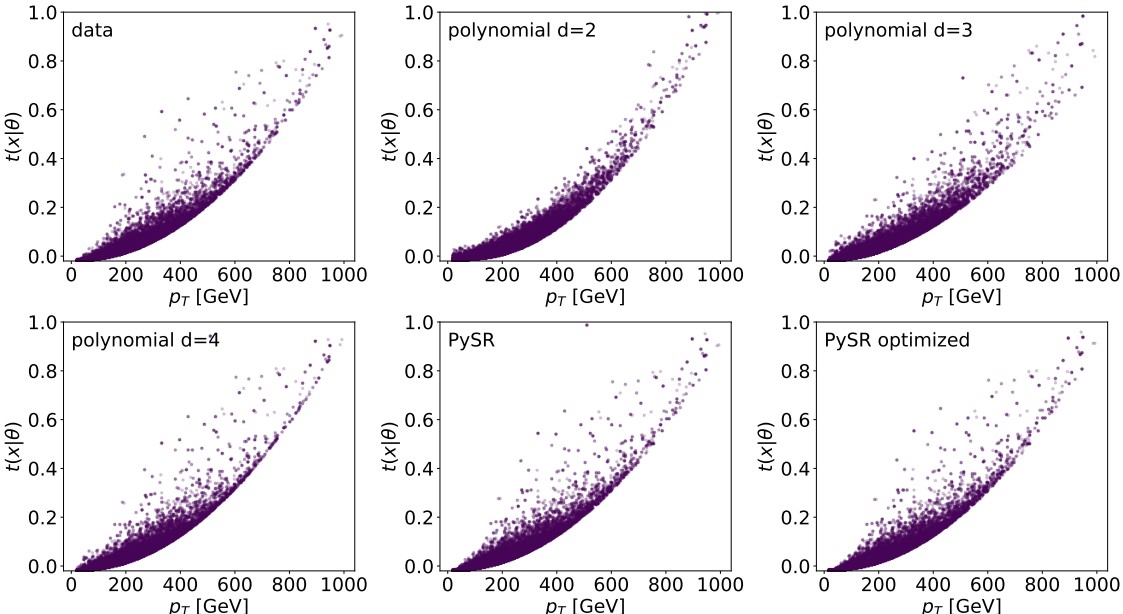

Figure 4: Score as a function of $p_T$ for the polynomial fits and the PySR output, including the optimization fit, for the simplified $ZH$ setup with $f_B = 0$, corresponding to Tab. 2.

larger Wilson coefficients, the score indeed decreases approximately like $1/\theta \sim 1/f_B$. For $\eta_-$ and in the boosted regime we see the same pattern, namely that the score decreases when we evaluate it away from the Standard Model. For all values of $f_B$ the score increases towards larger $\eta_-$, where events are generally more rare.

## 3.2 Learning a score formula

Now that we have a numerical definition of the score over phase space, we can use symbolic regression to construct a formula for its phase space distribution. From our earlier consideration we expect the score to be described by the two observables $p_T$ and $\eta_-$. Moreover, from Fig. 3 we expect that for small $f_B$ values the score should be covered by a polynomial in the leading observables $p_T/m_H$ and $\eta_-$.

**Polynomial functions for $f_B = 0$**

As a starting point, we extract a functional form of the score for $ZH$ production only including the $Z$-contribution and one quark flavor using a polynomial form in

$$x_p = \frac{p_T}{m_H}, \qquad \text{and} \qquad x_\eta = |\eta_-|. \tag{23}$$

The scaling ensures that all involved quantities are in the same order of magnitude which is easier for PySR to deal with. These phase space variables do not directly correspond to the variables in Eq.(22), but will allow us to generalize our results to the full hadron collider kinematics.

In the upper left panel of Fig. 4 we first show the full data set for $t(x|f_B = 0)$ as a function of $p_T$. Before applying PySR, we first establish a baseline by fitting polynomials of degrees two to four in $x_\eta$ and $x_p$. The fits minimized the MSE for all 500k phase space points. For the fits as well as for the optimizations of PySR results described below we use the python package LMFIT [48] for non-linear optimization and curve fitting which is based on SCIPY.OPTIMIZE [49].

In Tab. 2 we see that the increased expressivity of the higher polynomial leads to a slight improvement in the MSE value. From the prefactors we get a rough idea what the leading dependences are. According to the upper row of Fig. 4 the second-order polynomial describes most of the data well. The quadratic form, with four prefactors of similar size and a much smaller constant and $x_\eta^2$ term, is necessary to add the scattered points with large score at intermediate $p_T$-values and large $|\eta_-|$. This pattern reflects the fact that the score function for our toy model at $f_B = 0$, shown approximately in Eq.(22), is easy to model.

PYSR with the settings described in Sec. 2.3 with 10 populations and the maximal complexity of 50 gives us a hall of fame with the most prominent formulas listed in Tab. 3. The complexity refers to the original PYSR tree and can often be smaller when we simplify the equation by hand. The great advantage of PYSR is that given such a hall of fame we can choose a result that fits our needs best in terms of balancing complexity versus MSE. The last expression with complexity 49 corresponds to the PYSR result given in Tab. 2. It includes powers up to $p_T^2 |\eta_-|^3$, but leaves out some of the terms, notably $p_T^3$ and $p_T^4$, which are also missing from Eq.(22). Instead, PYSR introduces correlations between $p_T$ and $\eta_-$ to model their dependence. Overall, we see that while having less free parameters it gives better results than the polynomial of degree four.

An algorithmic weakness of PYSR is that it never properly fits its functional form to the data set. Because larger data sets pose an increasing challenge to PYSR we only use 800 of our originally 500k data points, distributed appropriately. For both reasons, we add an optimization fit for all parameters in the HoF functions using the whole data set. The shift in the parameters is indicated in the right column of Tab. 2, where the individual parameters change by up to a factor 2, and the error bar of the fit indicates that the original PYSR choice it outside the fit uncertainty. The modest improvement in the description of the score as a function of $p_T$ is illustrated in Fig. 4. The results given in Tab. 3 are also optimized.

Table 2: Polynomial score functions for the simplified $ZH$ setup with $f_B = 0$. The right column shows the results from an optimization fit to the PYSR function. For numerical reasons all results describe $t(x_p, x_\eta) \times 10$.

| | polynomial $d = 2$ | polynomial $d = 3$ | polynomial $d = 4$ | PYSR | PYSR optimized |
|---|---|---|---|---|---|
| MSE | $3.49 \cdot 10^{-3}$ | $8.16 \cdot 10^{-4}$ | $1.28 \cdot 10^{-4}$ | $1.23 \cdot 10^{-4}$ | $7.65 \cdot 10^{-5}$ |
| dof | 6 | 10 | 15 | 9 | 9 |
| 1 | -0.03145 | -0.1810 | -0.1231 | -0.1495 | -0.134807(46) |
| $x_p$ | -0.2022 | 0.4871 | -0.06404 | -0.01553 | -0.036030(78) |
| $x_\eta$ | -0.1783 | 0.1837 | -0.04830 | 0.0045 | 0.002083(55) |
| $x_p^2$ | 0.1805 | 0.1303 | 0.1612 | 0.1453 | 0.148277(26) |
| $x_p x_\eta$ | 0.2303 | -0.3434 | 0.1124 | -0.01553 | -0.00787(10) |
| $x_\eta^2$ | 0.02861 | -0.1036 | 0.06492 | - | - |
| $x_p^3$ | - | -0.001788 | $-4.504 \cdot 10^{-4}$ | - | - |
| $x_p^2 x_\eta$ | - | 0.1022 | -0.03152 | 0.01854 | 0.022835(68) |
| $x_p x_\eta^2$ | - | 0.1449 | -0.1551 | - | - |
| $x_\eta^3$ | - | 0.01001 | -0.01976 | $6.333 \cdot 10^{-4}$ | 0.0013648(50) |
| $x_p^4$ | - | - | $6.936 \cdot 10^{-5}$ | - | - |
| $x_p^3 x_\eta$ | - | - | -0.002264 | - | - |
| $x_p^2 x_\eta^2$ | - | - | 0.07835 | 0.005143 | -0.002813(67) |
| $x_p x_\eta^3$ | - | - | 0.03080 | -0.007064 | -0.011333(26) |
| $x_\eta^4$ | - | - | 0.001368 | - | - |
| $x_p^2 x_\eta^3$ | - | - | - | 0.01970 | 0.023525(22) |

Table 3: Score hall of fame for simplified $ZH$ setup with $f_B = 0$. The last formula corresponds to the PYSR result shown in Tab. 2. For numerical reasons all results describe $t(x_p, x_\eta) \times 10$.

| cmpl | dof | function | MSE |
|------|-----|----------|-----|
| 7 | 1 | $ax_p(x_p + x_\eta)$ | $3.81 \cdot 10^{-2}$ |
| 10 | 3 | $ax_p^2(b + x_\eta) - c$ | $2.49 \cdot 10^{-3}$ |
| 14 | 3 | $ax_p^2 + bx_p^2 x_\eta^2 - c$ | $6.64 \cdot 10^{-4}$ |
| 22 | 4 | $ax_p^2 + bx_p^2 x_\eta^2 - cx_p x_\eta - d$ | $3.09 \cdot 10^{-4}$ |
| 32 | 6 | $a(x_p^2 + x_\eta) + bx_p^2 x_\eta - (cx_p - d)^2 + ex_p^2 x_\eta^3 - f$ | $2.06 \cdot 10^{-4}$ |
| 34 | 7 | $a(x_p^2 + x_\eta) + bx_p^2 x_\eta - (cx_p - d)^2 + ex_\eta^3 (x_p - f)^2 - g$ | $7.77 \cdot 10^{-5}$ |
| 49 | 9 | $ax_p^2 + bx_p^2 x_\eta - cx_\eta (x_p - d) + ex_\eta^3 (x_p - f)^2 + gx_p^2 x_\eta^2 - hx_p - i$ | $7.65 \cdot 10^{-5}$ |

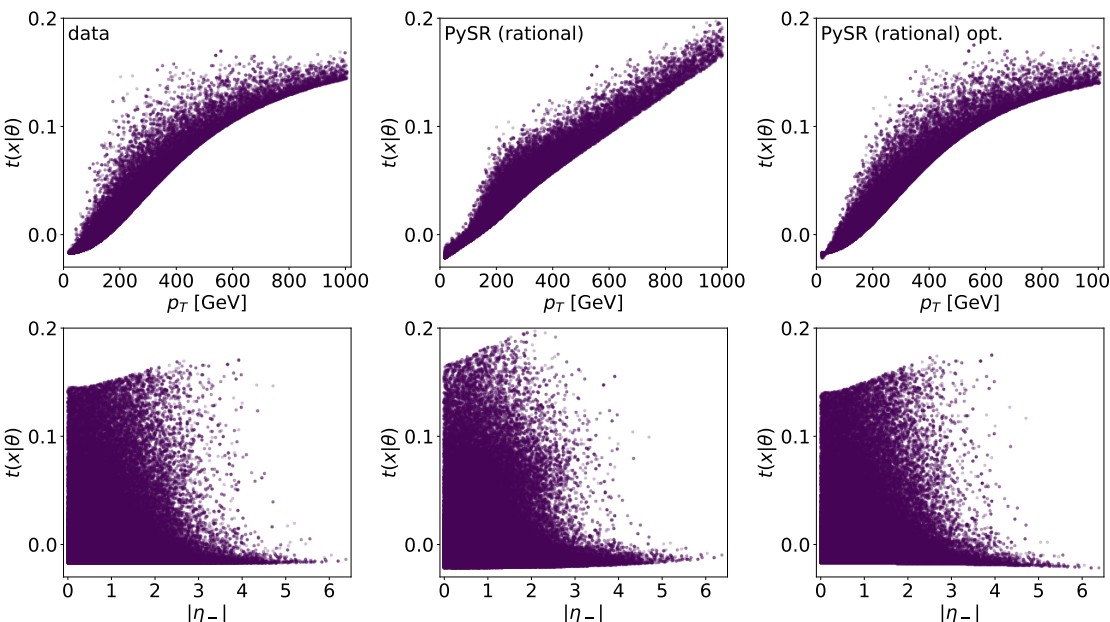

Figure 5: Score as a function of $p_T$ and $\eta_-$ for the rational PYSR output for the simplified $ZH$ setup with $f_B = 10$, corresponding to Tab. 4.

**Rational function for $f_B = 10$**

Moving on to a more challenging PYSR task, we know from Tab. 1 and Fig. 3 that a simple polynomial form is unlikely to describe the score away from the Standard Model, for instance at $f_B = 10$. To enable PYSR to describe this score, we also allow for the division operator, so the score can be described by a rational function. The maximum complexity is now 75.

The initial PYSR output we chose from the hall of fame is the function

$$t(x_p, x_\eta | f_B = 10) = ax_p - b + \cfrac{c(x_\eta + d)}{e + \cfrac{f}{x_p\left(\left(x_p - g\right)^4 + h\right)\left(i\left(x_\eta - j\right)^2 + k\right)}}, \qquad (24)$$

again with $x_p = p_T / m_H$ and $x_\eta = |\eta_-|$. In Tab. 4 we see that with this formula PYSR initially finds stable results, but a proper fit converges on some very large parameters with large error bars, specifically $c$, $e$ and $f$. This reflects flat directions in Eq.(24), which we can remove by

re-defining

$$t(x_p, x_\eta | f_B = 10) = ax_p - b + \cfrac{c'x_\eta + d'}{1 + \cfrac{f'}{x_p \left( h' \left( x_p + g \right)^4 - 1 \right) \left( i' \left( x_\eta - j \right)^2 + 1 \right)}}, \qquad (25)$$

with

$$c' = \frac{c}{e}, \qquad d' = \frac{cd}{e}, \qquad f' = \frac{f}{ehk}, \qquad h' = \frac{1}{h}, \qquad i' = \frac{i}{k}. \qquad (26)$$

This way we remove two parameters, $e$ and $k$. We see in the third column of Tab. 4 that now all parameters come with controlled uncertainties.

Finally, we can check if the two exponents in the function are what they should be. The final function we can fit to our data set is then

$$t(x_p, x_\eta | f_B = 10) = ax_p - b + \cfrac{c'x_\eta + d'}{1 + \cfrac{f'}{x_p \left( h' |x_p + g|^z - 1 \right) \left( i' |x_\eta - j|^y + 1 \right)}}. \qquad (27)$$

According to the right column of Tab. 4, this leads to a sizeable shift in one of the exponents, $z = 2 \to 3.37$. On the other hand, from the very slight improvement in the MSE we see that already the original function was expressive enough to describe the majority of data points.

In Fig. 5 we show the dependence of the rational score functions on the two kinematic observables. Here we see that the post-processing is necessary to describe the high-$p_T$ range, as well as the $|\eta_-|$-dependent upper limit. Given that in an actual analysis we rely on parameter points with large score to measure $f_B$, such a difference might become numerically relevant. We will come back to the relation between MSE and analysis reach in Sec. 4.4.

Table 4: Rational score parametrizations for the simplified $ZH$ setup with $f_B = 10$. We show parameters from PySR, from an additional fit to the PySR function, and from a fit including exponents. For numerical reasons all results describe $t(x_p, x_\eta) \times 10$.

|  | PySR default | PySR optimized | | |
|---|---|---|---|---|
|  |  | Eq.(24) | Eq.(25) | Eq.(27) |
| MSE | $8.85 \cdot 10^{-4}$ | $7.52 \cdot 10^{-5}$ | $7.38 \cdot 10^{-5}$ | $5.42 \cdot 10^{-5}$ |
| $a$ | 0.2201 | 0.02318(20) | 0.01534(20) | 0.00805(17) |
| $b$ | 0.2427 | 0.169067(79) | 0.166262(71) | 0.166229(67) |
| $c^{(\prime)}$ | 0.0249 | 6.2(10) | 0.09973(32) | 0.06691(36) |
| $d^{(\prime)}$ | 0.7070 | 13.667(54) | 1.5949(23) | 1.712(22) |
| $e$ | 0.1405 | 56.6(96) | - | - |
| $f^{(\prime)}$ | 0.7046 | 374(42) | 2.680(21) | 1.928(18) |
| $g$ | 0.2855 | -13.834(86) | 18.56(14) | 23.54(20) |
| $h^{(\prime)}$ | 0.1270 | $-3.945(96) \cdot 10^4$ | $7.97(23) \cdot 10^{-6}$ | $1.206(33) \cdot 10^{-5}$ |
| $i^{(\prime)}$ | 0.5750 | $2.05(30) \cdot 10^{-5}$ | 0.42702(54) | 0.05091(78) |
| $j$ | 0.3189 | 0.336749(58) | 0.32375(55) | -0.5942(55) |
| $k$ | 0.1192 | $4.61(67) \cdot 10^{-5}$ | - | - |
| $y$ | fixed 2 | fixed 2 | fixed 2 | 3.3771(78) |
| $z$ | fixed 4 | fixed 4 | fixed 4 | 3.5724(43) |

**Including photon for $f_B = 10$**

In our next step we add the $s$-channel photon to the process and study how an increased complexity helps describing the score for $f_B = 10$. It turns out that the default setup of PYSR does not find a good high-complexity function for this case, because the algorithm gets stuck at complexities around 30. The reason for this problem is the mutation probability Eq.(12), which for small parsimony reads

$$p = \exp\left(-\frac{\text{MSE}_{\text{new}} - \text{MSE}_{\text{old}}}{\texttt{alpha} \cdot T \cdot \texttt{baseline}}\right). \tag{28}$$

The baseline is an order-one constant. This form causes a problem if the old function is a poor fit, and the new function has an improved shape, but an even worse MSE for its initial parameters. In that case the absolute scale of the MSE values always leads to a vanishing mutation probability, and Eq.(12) or Eq.(28) do not accept enough more complex functions to leave the local minimum. Shifting `alpha` to very large values helps, but leads to problems when the typical MSE become small. For data that is easy to describe, as our previously considered cases, this problem was compensated by a very large number of mutation attempts, but after including the photon this compensation fails.

Once we understand the problem, it is easy to fix with a new mutation probability,

$$p = \exp\left(-\frac{\text{MSE}_{\text{new}} - \text{MSE}_{\text{old}}}{\texttt{alpha} \cdot T \cdot \text{MSE}_{\text{old}}}\right). \tag{29}$$

In the following we use this relative difference with `alpha = 100`.

For two $s$-channel diagrams and $f_B = 10$ we show a selection of the HoF functions in Tab. 5. As expected, PYSR produces results with larger complexities, driven by an MSE improvement by two orders of magnitude. We illustrate the improved MSE with increased complexity in the left panel of Fig. 6. After removing flat directions, the best-suited rational function in the HoF retains 11 parameters and reads

$$t(x_p, x_\eta | f_B = 10) = \cfrac{x_p - a}{bx_p^3 + \cfrac{cx_p - \cfrac{d}{x_p}\left(x_\eta + e - \cfrac{f}{(x_p - g)^2(x_p + x_\eta^2) + h}\right)}{x_p + ix_\eta^2(x_p + j)} + 1} - k. \tag{30}$$

Table 5: Score hall of fame for the simplified $ZH$ setup with $f_B = 10$ and $s$-channel photon and $Z$. For numerical reasons all results describe $t(x_p, x_\eta) \times 10$.

| cmpl | dof | function | MSE |
|------|-----|----------|-----|
| 16 | 5 | $ax + by - c(d - ex)^2$ | $1.57 \cdot 10^{-2}$ |
| 22 | 6 | $ax + by - c(d - ex)^2 + f/x$ | $9.46 \cdot 10^{-3}$ |
| 30 | 8 | $(ax - b)/(cx^3 + d + e(x - y + f + g/x)/x) - h$ | $3.82 \cdot 10^{-3}$ |
| 42 | 9 | $(ax - b)/(cx^3 + d + e(x + f - (gy - h/x^2)/x)/(x + y/x)) - i$ | $1.22 \cdot 10^{-3}$ |
| 45 | 8 | $(x - a)/(bx^3 + c + d(x + e - f(y - g/x^2(x + y))/x)/(x + y^2/x)) - h$ | $7.96 \cdot 10^{-4}$ |
| 47 | 10 | $(x - a)/(bx^3 + c + d(x + e - f(y - g/(hx^2(x + y) + i))/x)/(x + y^2/x)) - j$ | $6.71 \cdot 10^{-4}$ |
| 50 | 10 | $(x - a)/(bx^3 + c + d(x + e - f(y - g/(hx^2(x + y^2 - y) + i))/x)$ $/(x + y^2/x)) - j$ | $6.03 \cdot 10^{-4}$ |
| 63 | 13 | $(ax - b)/(cx^3 + d + e(x + f - g(hx^2 + y + i - j$ $/(kx^2(x + (y - l)^2 - y) + m))/x)/(x + y^2/x)) - n$ | $5.64 \cdot 10^{-4}$ |
| 73 | 14 | $(ax - b)/(cx^3 + d(x - e(fx^2 + y + g - h/(i(j - x)^2(x + y^2) + k))/x)$ $/(x + ly(mxy + y)) + n) - o$ | $1.45 \cdot 10^{-4}$ |

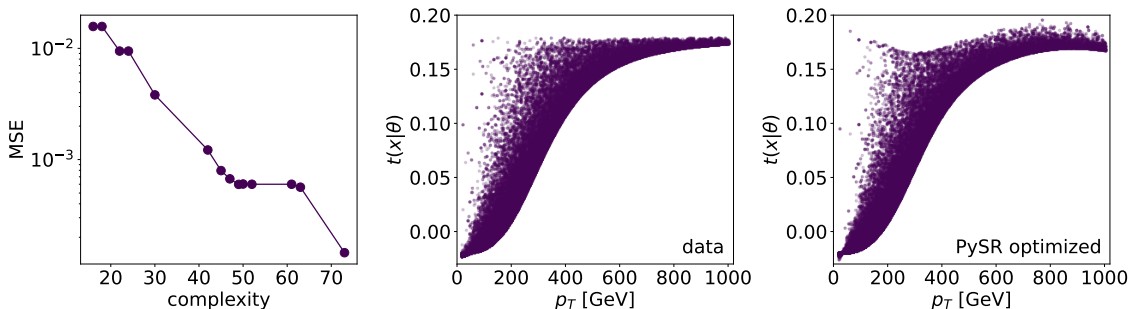

Figure 6: MSE and score for the simplified $ZH$ setup with $f_B = 10$ and $s$-channel photon and $Z$. The functional forms correspond to Tab. 5. MSE given for $t(x_p, x_\eta) \times 10$.

In spite of the large complexity, this function does still not describe the score perfectly. In the center and right panels of Fig. 6 we see that points close to the upper score limit and points at large $p_T$ still show deviations from the training data.

## 3.3 Two quark flavors

Finally, we need to include different incoming quark flavors for

$$pp \rightarrow ZH, \tag{31}$$

as an example for an unobserved or unphysical parameter in the joint score in Eq.(6), which we remove to arrive at the physical score or optimal observable.

**Results for $f_B = 0$**

In Tab. 6 we show a set of function from the HoF with their corresponding MSE for the Standard Model parameter choice $f_B = 0$. We remind ourselves that in this case the functional form will most likely be described by a simple polynomial in $x_p = p_T/m_H$ and $x_\eta = |\eta_-|$. Increasing the complexity from 7 to 29, or the number of degrees of freedom from one to eight has a surprisingly mild effect on the MSE. We can understand the reason when looking at the kinematic distribution of the score in Fig. 7. In the left panel we see that integrating out the discrete quark flavor leads to two distinct branches in the score, an upper branch for incoming $d$-quarks and a lower branch for incoming $u$-quarks. Because the information is unphysical,

Table 6: Score hall of fame for the complete $ZH$ setup with $f_B = 0$. For numerical reasons all results describe $t(x_p, x_\eta) \times 10$.

| cmpl | dof | function | | MSE |
|---|---|---|---|---|
| 7 | 1 | $ax_p(x_p + x_\eta)$ | $a = 0.0375$ | $6.51 \cdot 10^{-3}$ |
| 9 | 2 | $ax_p^2(x_\eta + b)$ | $a = 0.0203 \; ab = 0.0406$ | $4.35 \cdot 10^{-3}$ |
| 11 | 2 | $ax_p^2(x_\eta^2 + b)$ | $a = 0.0111 \; ab = 0.0462$ | $4.32 \cdot 10^{-3}$ |
| 13 | 3 | $ax_p^2 + bx_px_\eta^2 - c$ | $a = 0.0648 \; b = 0.0088 \; c = 0.0625$ | $1.96 \cdot 10^{-3}$ |
| 17 | 4 | $ax_p^2 + bx_px_\eta^2 - cx_\eta - d$ | | $1.84 \cdot 10^{-3}$ |
| 19 | 4 | $ax_p^2 + bx_px_\eta^2 - cx_p - dx_\eta$ | | $1.74 \cdot 10^{-3}$ |
| 21 | 5 | $ax_p^2 + bx_px_\eta^2 - cx_p - dx_\eta + e$ | | $1.72 \cdot 10^{-3}$ |
| 27 | 6 | $ax_p^2 + bx_px_\eta^2 - cx_px_\eta - dx_p + ex_\eta + f$ | | $1.63 \cdot 10^{-3}$ |
| 28 | 7 | $ax_p^2(bx_\eta - c)^2 + dx_px_\eta^2 + ex_p^2 - fx_p - gx_\eta$ | | $1.43 \cdot 10^{-3}$ |
| 29 | 8 | $ax_p^2 + b(x_\eta^2 + c)(x_\eta(dx_p - e)(x_p - f) + x_p + g) - h$ | | $1.29 \cdot 10^{-3}$ |

Table 7: Score hall of fame for the complete $ZH$ setup with $f_B = 10$. For numerical reasons all results describe $t(x_p, x_\eta) \times 10$.

| cmpl | dof | function | | MSE |
|------|-----|----------|---|-----|
| 10 | 3 | $ax_p + bx_\eta^3 - c$ | $a = 0.3487\ b = 0.0043\ c = 0.3492$ | $1.61 \cdot 10^{-2}$ |
| 16 | 4 | $ax_p - b/(cx_p^4 x_\eta + d)$ | $a = 0.3032\ b = 0.0960\ c = 0.0213\ d = 0.3033$ | $1.26 \cdot 10^{-2}$ |
| 20 | 4 | $ax_p - b/(cx_p^5 x_\eta + d)$ | $a = 0.2860\ b = 0.0942\ c = 0.0117\ d = 0.3005$ | $1.21 \cdot 10^{-2}$ |
| 23 | 5 | $ax_p + bx_\eta^3 - c/(dx_p^4 x_\eta + e)$ | | $1.19 \cdot 10^{-2}$ |
| 25 | 7 | $ax_p + bx_\eta^3 + cx_\eta - d/(ex_p^4(x_\eta + f) + g)$ | | $1.14 \cdot 10^{-2}$ |
| 45 | 12 | $ax_p + bx_\eta - c(x_p - d)^3 + e - f/(gx_p^3 x_\eta^3 - x_\eta(hx_p + i) + j(x_p + k)^6 + l)$ | | $4.65 \cdot 10^{-3}$ |
| 51 | 13 | $ax_p + bx_\eta - c(x_p - d)^3 + e - f/(gx_p^3 x_\eta^3 - x_\eta(h + i) + j(x_p + k)^6 + l + m/x_p)$ | | $4.65 \cdot 10^{-3}$ |

an implicit or explicit form for the score will interpolate between them and define a single curve in the middle with an MSE well above the case without unphysical parameters shown in Tab. 2.

The simplest expression of complexity seven consists of a squared term in $p_T$ and a linear correlation of $p_T$ and $|\eta_-|$. It describes the data for small $p_T$ but undershoots for larger values. More importantly, its $|\eta_-|$-dependence is simply too flat. Nevertheless, already this simple form describes most of the data points at low $p_T$ and central $|\eta_-|$. Switching to a squared correlation term with complexity 11 leads to a slight improvement in the $\eta_-$ distribution for low $p_T$, but still does not give the correct shape at large $p_T$. Interestingly, another slight complexity increase to 13 improves the description at large $p_T$, but worsens it at large $\eta_-$, indicating a tension for a limited number of parameters.

Eventually, moving towards an appropriate complexity we see that PYSR starts adding linear terms in $p_T$ and $|\eta_-|$, which slightly improves the MSE in the bulk of central events with small $p_T$, but still does not fit the data points with large scores. This situation changes for complexity with terms proportional to $p_T^3$ and $|\eta_-|^3$, including a more complex set of correlations between them. This is consistent with the results for our toy model in Tab. 2, and we find that adding more complexity does not improve the MSE further.

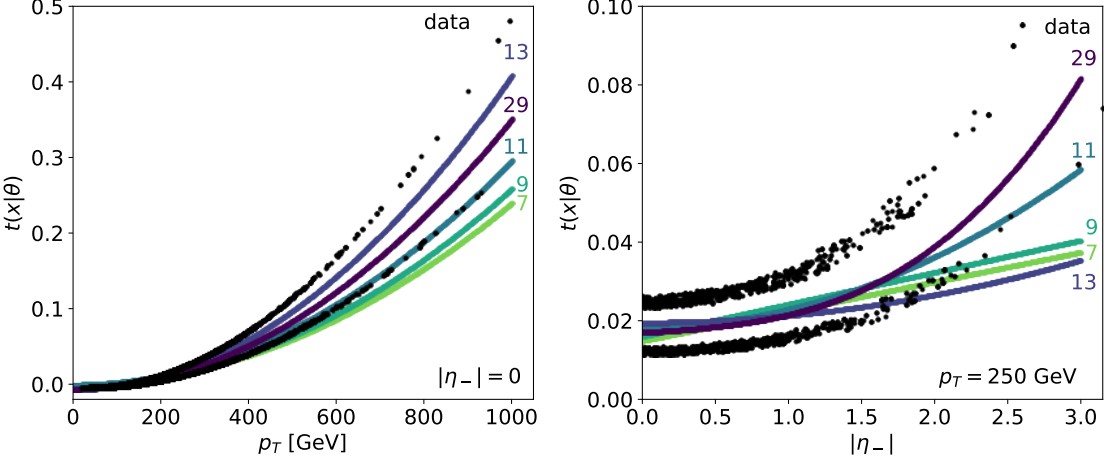

Figure 7: Sliced kinematic distributions for the joint score in the complete $ZH$ setup with $f_B = 0$, showing the HoF given in Tab. 6.

**Results for $f_B = 10$**

Finally, we can see what kind of rational function PySR constructs for the full $ZH$ process with $f_B = 10$. In Tab. 7 we see that the simplest solution at complexity 10 already uses three parameters, and Fig. 8 confirms that it is does not provide a good interpolation between the two branches. With increasing complexity, all formulas up to complexity 45 only include a linear $p_T$-term in the numerator and therefore fail to describe the intermediate $p_T$-range and the saturation above. Note that for high complexity the denominator includes powers up to $p_T^6$ to describe the rapid saturation. At the same time, a $p_T^3$-term in the numerator allows the function to describe the low- and intermediate-$p_T$ range well. As for $f_B = 0$, adding more complexity does not improve the MSE, which is now limited by the interpolation between the two branches. The slight over-shoot for large $|\eta_-|$ affects a too small fraction of parameter points to make a difference.

Our extensive discussion of the simple $ZH$ production process shows that PySR can extract useful analytic expressions for the score or the optimal observable. This can be simple polynomials — which could also be extracted through a simple fit — or rational functions, for which a general parametrization would lead to a very large number of parameters. For the case without unphysical parameters we can improve the MSE with increasing complexity, while for the case of two incoming quark flavors we see that the achievable MSE is limited, and adding complexity to the score stops improving the result. For the two questions, namely if PySR finds the *correct* score or optimal observable and how the PySR result performs in setting limits in an LHC analysis we turn to the better-understood example of $CP$-violation in weak boson Higgs production.

# 4 WBF Higgs production and CP

Going beyond our simple toy scenario, we can apply the same methodology to the more complex WBF Higgs production process and the fundamentally interesting question of $CP$-violation in the $VVH$ interaction. For this case we know the form of the optimal observable at parton level and close to the Standard Model, so we can check if PySR extracts the correct score, what changes when we include detector effects, and what kind of reach we can expect from different functional forms.

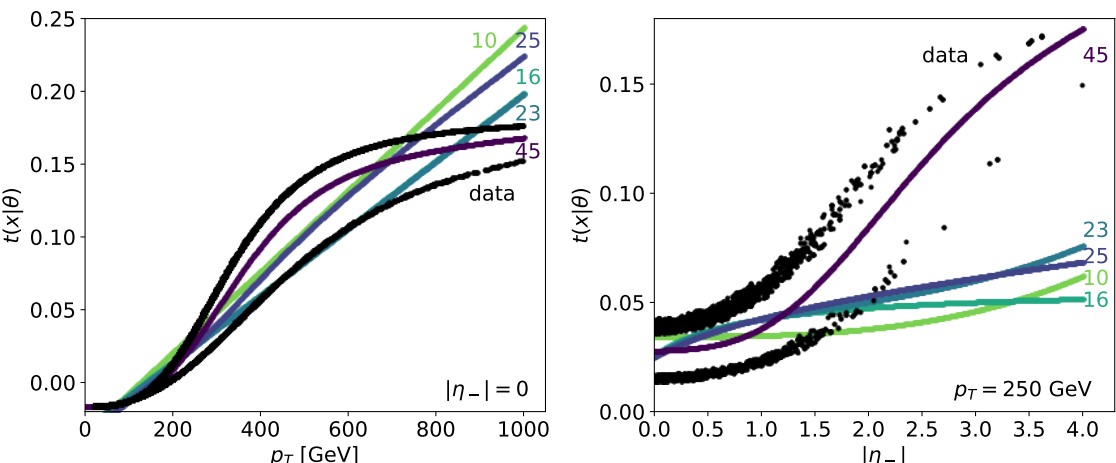

Figure 8: Sliced kinematic distributions for the complete $ZH$ setup with $f_B = 10$, showing the HoF given in Tab. 7.

## 4.1 Score for $f_{W\widetilde{W}}$

Testing the properties of the $VVH$ vertex in WBF Higgs production

$$pp \to Hjj, \qquad \text{with} \qquad |\mathcal{M}|^2 \propto \alpha^2, \tag{32}$$

is equivalent to corresponding analyses of $VH$ production and $H \to VV$ decays, with the advantage that we do not have to rely on a precise reconstruction of the Higgs decay products [21, 50]. We also know that the signed azimuthal angle between the tagging jets $\Delta\phi$ [21, 29, 30] is the appropriate genuine $CP$-odd observable. To define an optimal observable we choose the specific $CP$-violating operator

$$\mathcal{L} = \mathcal{L}_{\text{SM}} + \frac{f_{W\widetilde{W}}}{\Lambda^2} \mathcal{O}_{W\widetilde{W}}, \qquad \text{with} \qquad \mathcal{O}_{W\widetilde{W}} = -(\phi^\dagger \phi)\,\widetilde{W}^k_{\mu\nu} W^{\mu\nu k}. \tag{33}$$

For our numerical results we quote $f_{W\widetilde{W}}$-values for $\Lambda = 1$ TeV. In Fig. 9 we show the effect of this additional operator on the WBF kinematics. First, $\Delta\phi$ develops an asymmetric form, which can most easily be exploited through an asymmetry measurement. Second, the higher-dimensional operator $\mathcal{O}_{W\widetilde{W}}$ with its additional momentum dependence induces a harder tagging jet spectrum, an effect which it shares with many other higher-dimensional operator, and which is not related to $CP$-violation. On the other hand, there exist no dimension-4 operators leading to $CP$ violation in the $VVH$ interaction, so when we search for the leading effect from $\mathcal{O}_{W\widetilde{W}}$ this momentum dependence will enhance the LHC reach.

For the leading partonic contribution from $WW$-fusion,

$$ud \to Hdu, \tag{34}$$

with the standard tagging jet cuts $|\eta_j| < 5$, $|\Delta\eta_{jj}| > 2$, and $p_{T,j} > 20$ GeV we can compute the score contribution given in Eq.(18) for the Standard Model point $f_{W\widetilde{W}} = 0$ and find [21]

$$t(x|f_{W\widetilde{W}} = 0) \approx -\frac{8v^2}{m_W^2} \frac{(k_d k_u) + (p_u p_d)}{(p_d p_u)(k_u k_d)}\, \epsilon_{\mu\nu\rho\sigma}\, k_d^\mu k_u^\nu p_d^\rho p_u^\sigma, \tag{35}$$

where $k_{u,d}$ are the incoming and $p_{u,d}$ the outgoing quark momenta. We can relate this form to $\Delta\phi$ when we assign the incoming momenta to a positive and negative hemisphere, $k_\pm = (E_\pm, 0, 0, \pm E_\pm)$ and correspondingly for the outgoing momenta $p_\pm$. We then find

$$t(x|f_{W\widetilde{W}} = 0) \approx -\frac{8v^2}{m_W^2} \frac{2E_+ E_- + (p_+ p_-)}{(p_+ p_-)} p_{T+} p_{T-}\, \sin\Delta\phi, \tag{36}$$

with the known dependence $t \propto \sin\Delta\phi$. The momentum-dependent prefactor reflects the dimension-6 structure with an approximate scaling $t \propto p_{T+} p_{T-}$.

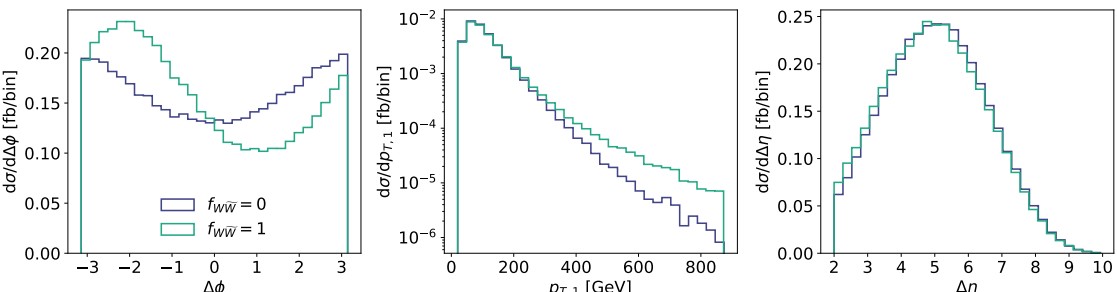

Figure 9: Kinematic distributions for WBF Higgs production at parton level with different Wilson coefficients $f_{W\widetilde{W}}$. Here, $\Delta\phi$ denotes the signed azimuthal angle between the two tagging jets, $p_{T,1}$ refers to the leading tagging jet, and $\Delta\eta = |\Delta\eta_{jj}|$.

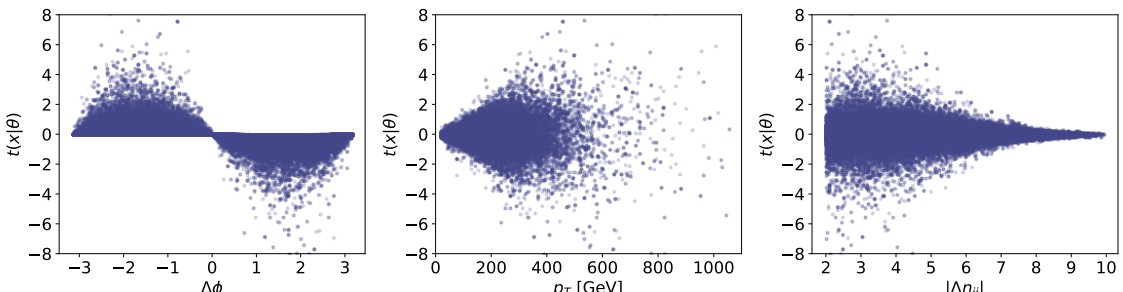

Figure 10: Score for simplified WBF Higgs production at parton level and with $f_{W\widetilde{W}} = 0$.

## 4.2 Symbolic regression at parton level

As before, we first use symbolic regression on the simplified partonic process

$$ud \to Hjj, \qquad (37)$$

without shower or detector effects. For this setup we will extract the score for the Standard Model parameter point $f_{W\widetilde{W}} = 0$ and for $f_{W\widetilde{W}} = 1$. In Fig. 9 we see that for $f_{W\widetilde{W}} = 0$ the $\Delta\phi$ distribution is symmetric, while for $f_{W\widetilde{W}} = 1$ it roughly follows a sine shape. The $p_{T,j}$-distribution indicates that for the two choices of reference point, the score formula will chance its momentum dependence.

### Results for $f_{W\widetilde{W}} = 0$

For small deviations from the $CP$-conserving Standard Model we show the score distributions in Fig. 10. Comparing the different kinematic observables, the leading dependence is clearly on $\Delta\phi$. Switching on $f_{W\widetilde{W}} > 0$ moves events from $\Delta\phi > 0$ to $\Delta\phi < 0$, as expected from Fig. 9. The actual shape of $t(\Delta\phi|f_{W\widetilde{W}})$ confirms the $\sin\Delta\phi$ scaling of Eq.(36). The dependence on $p_{T,1}$ indicates large absolute values of the score for harder events, which will boost the analysis when correlated with $\Delta\phi$. The dependence on $\Delta\eta = |\Delta\eta_{jj}|$ is comparably mild, so we expect PYSR to only add the tagging jet rapidities at high complexity.

To encode the score dependence of Fig. 10 we use PYSR on the observables

$$\left\{ x_{p,1}, x_{p,2}, \Delta\phi, \Delta\eta \right\}, \qquad \text{with} \qquad x_{p,j} = \frac{p_{T,j}}{m_H}, \qquad (38)$$

Table 8: Score hall of fame for simplified WBF Higgs production with $f_{W\widetilde{W}} = 0$, including a optimization fit.

| compl | dof | function | MSE |
|---:|---:|---|---:|
| 3 | 1 | $a\,\Delta\phi$ | $1.30 \cdot 10^{-1}$ |
| 4 | 1 | $\sin(a\Delta\phi)$ | $2.75 \cdot 10^{-1}$ |
| 5 | 1 | $a\Delta\phi x_{p,1}$ | $9.93 \cdot 10^{-2}$ |
| 6 | 1 | $-x_{p,1}\sin(\Delta\phi + a)$ | $1.90 \cdot 10^{-1}$ |
| 7 | 1 | $(-x_{p,1} - a)\sin(\sin(\Delta\phi))$ | $5.63 \cdot 10^{-2}$ |
| 8 | 1 | $(a - x_{p,1})x_{p,2}\sin(\Delta\phi)$ | $1.61 \cdot 10^{-2}$ |
| 14 | 2 | $x_{p,1}(a\Delta\phi - \sin(\sin(\Delta\phi)))(x_{p,2} + b)$ | $1.44 \cdot 10^{-2}$ |
| 15 | 3 | $-(x_{p,2}(a\Delta\eta^2 + x_{p,1}) + b)\sin(\Delta\phi + c)$ | $1.30 \cdot 10^{-2}$ |
| 16 | 4 | $-x_{p,1}(a - b\Delta\eta)(x_{p,2} + c)\sin(\Delta\phi + d)$ | $8.50 \cdot 10^{-3}$ |
| 28 | 7 | $(x_{p,2} + a)(bx_{p,1}(c - \Delta\phi)$ $-x_{p,1}(d\Delta\eta + ex_{p,2} + f)\sin(\Delta\phi + g))$ | $8.18 \cdot 10^{-3}$ |

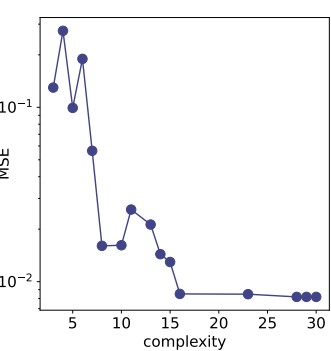

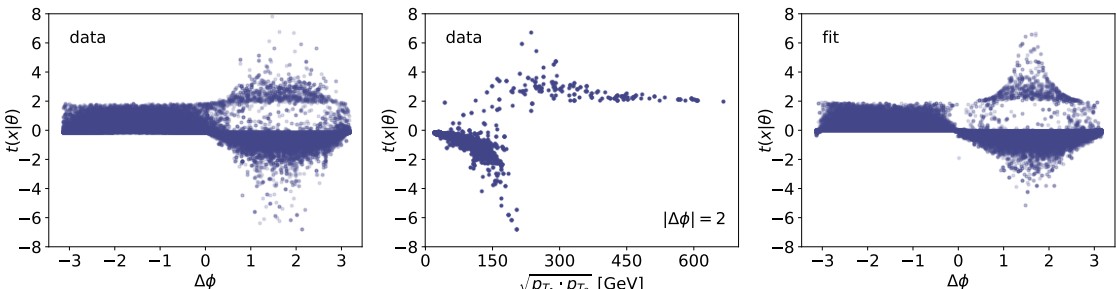

Figure 11: Score for simplified WBF Higgs production at parton level and with $f_{W\widetilde{W}} = 1$. The functional form for the right panel with complexity 31 is given in Tab. 9.

using the usual summing, subtraction, and multiplication operators and now adding the sine operator. These observables are inspired by our intuition about the physics of WBF processes. As a matter of fact, we do not expect the rapidities to be relevant for our CP-study, but we nevertheless include it as a check. Because symbolic regression is much more efficient in extracting an analytic form than for instance a polynomial fit it is no probem to include a relatively large set of observables just to ensure that they do not actually contribute to the final results.

We use the same PySR settings as in Sec. 2.3, except for `maxsize=30` and `alpha=1.5`. In Tab. 8 we show the results, alongside the improvement in the MSE. Starting with the leading dependence on $\Delta\phi$, PySR needs complexity 8 with one free parameter to derive $t \approx p_{T,1}p_{T,2}\sin\Delta\phi$. At this point it turns out that adding $\Delta\eta$ to the functional form still leads to a significant improvement with a 4-parameter description of complexity 16, namely

$$t(x_{p,1}, x_{p,2}, \Delta\phi, \Delta\eta | f_{W\widetilde{W}} = 0) = -x_{p,1}(x_{p,2} + c)(a - b\Delta\eta)\sin(\Delta\phi + d),$$
$$\text{with} \quad a = 1.086(11), \quad b = 0.10241(19), \quad c = 0.24165(20), \quad d = 0.00662(32). \quad (39)$$

The numbers in parentheses give the uncertainty from the optimization fit. Even though $d$ is significantly different from zero, it is sufficiently small that we can to first approximation neglect it and confirm the scaling $t \propto \sin\Delta\phi$. Similarly, the dependence on the rapidity difference $\Delta\eta$ is suppressed by $b/a \sim 0.1$. Beyond this point we do not find a significant improvement in the MSE relative to the true score.

## Results for $f_{W\widetilde{W}} = 1$

From previous cases we expect that moving away from the Standard Model will lead to a more complex score formula than Eq.(36). In Fig. 11 we show the score as a function of kinematic observables for $f_{W\widetilde{W}} = 1$. Comparing this $\Delta\phi$-dependence to Fig. 10 confirms that the simple scaling with $\sin\Delta\phi$ has indeed vanished. Instead, we observe an upper limit $t < 2$ for negative $\Delta\phi$, which according to Tab. 1 reflects the dominance of the positive, quadratic term with a scaling $t \sim 2/f_{W\widetilde{W}}$. The also positive contribution from the interference term remains numerically subleading.

For positive $\Delta\phi$ we observe a more complex pattern from the interplay of linear and quadratic contributions. The interference term still follows an anti-symmetric $\sin\Delta\phi$ shape and contributes negative scores for positive $\Delta\phi$. We can split the events into three phase space regions: interference-dominated with $t < 0$, quadratic-dominated with $t = 0 \ldots 2/f_{W\widetilde{W}}$, and again interference-dominated with $t > 2/f_{W\widetilde{W}}$. These regions can be separated through their $p_T$-dependence, shown in the center panels of Fig. 11. For small transverse momenta the interference with the dimension-6 contribution gives mostly negative scores, followed by an

Table 9: Score hall of fame for WBF Higgs production with $f_{W\widetilde{W}} = 1$.

| cmpl | dof | function | MSE |
|------|-----|----------|-----|
| 3  | 1 | $ax_{p,\times}$ | 0.124 |
| 12 | 2 | $ax_{p,\times}/(x_{p,\times}/\Delta\eta + \Delta\eta + b)$ | 0.116 |
| 15 | 2 | $(s_\phi + a)(-s_\phi + x_{p,\times} - b)/(-s_\phi + x_{p,\times} + \Delta\eta/x_{p,\times})$ | 0.054 |
| 26 | 4 | $a/(b - (s_\phi - c - d/(s_\phi^2 - s_\phi\Delta\eta - s_\phi/x_{p,\times} + ex_{p,\times}^2))/x_{p,\times})$ | 0.048 |
| 31 | 7 | $a/(b - (s_\phi + (cs_\phi^2 - d)/(es_\phi^2 x_{p,\times}^2 - s_\phi\Delta\eta + f) - g)/x_{p,\times})$ | 0.039 |

intermediate regime with a broad range of score values, until for large transverse momenta that score is concentrated at the limit $t = 2/f_{W\widetilde{W}} = 2$ from the quadratic contribution.

After confirming that turning the more complex phase space dependence for $f_{W\widetilde{W}} = 1$ into a formula will be challenging, we change the parameter basis to

$$\left\{ x_{p,\times} = \frac{\sqrt{p_{T,1}p_{T,2}}}{m_H}, \ s_\phi = \sin\Delta\phi, \ \Delta\eta \right\}, \tag{40}$$

and allow for summing, subtraction, multiplication, and division operators. Adding a second $p_T$-parameter like $p_{T,1} + p_{T,2}$ does not lead to a significant improvement. The corresponding HoF is shown in Tab. 9. First, we see that the MSE we can achieve is almost one order of magnitude worse than for $f_{W\widetilde{W}} = 0$. The 7-parameter form generated with complexity 31 can be written as the rational function

$$t(x_{p,\times}, s_\phi, \Delta\eta | f_{W\widetilde{W}} = 1) = \frac{a' x_{p,\times}(e' s_\phi^2 x_{p,\times} - s_\phi\Delta\eta - f')}{(b' x_{p,\times} + s_\phi - g')(e' s_\phi^2 x_{p,\times} - s_\phi\Delta\eta - f') - c' s_\phi^2 - d'},$$

with $\quad a' = 0.75, \quad b' = 0.38, \quad c' = 4.2, \quad d' = 4.6, \quad e' = 1.1, \quad f' = 0.26, \quad g' = 0.21.$
$$\tag{41}$$

As for the $ZH$ case with $f_B = 10$ the functional form is not particularly enlightening, aside from the fact that the rational form can generate the observed cutoff $t < 2/\theta$ for large Wilson coefficients and that it has nothing to do with the simple scaling $t \propto s_\phi$ for $f_{W\widetilde{W}} = 0$.

## 4.3 Detector effects

Given that all our results have been derived at parton level, the obvious question is what impact a detector simulation will have on our analytic expressions for the optimal observables. In this section we will use the same process, WBF Higgs production, but add parton shower and fast detector simulation with DELPHES [41] using the default CMS card including the anti-$k_t$ jet algorithm [51] implemented in FASTJET [52].

To avoid the additional complication of having to select the two forward jets, we do not allow for initial state radiation and postpone all question concerning final states with a flexible number of particles to a more detailed study. While virtual corrections implented for instance in Madgraph should not be a challenge to the extraction of an analytic optimal observable at all, real emission corrections or jet radiation would need to be accomodated in the choice of relevant observables and invariably lead to the question what the appropriate observables for describing the hard process are.

After including detector effects, MADMINER still extracts the joint score from parton level observables while for the fitting process we are limited to the final-state observables.

In general, detector effects will mostly add noise to the data, which we find to affect the PYSR convergence. For $f_{W\widetilde{W}} = 0$ we still find the same kind of expressions as without detector effects, for instance the 4-parameter expression given in Eq.(39). To estimate the detector

Table 10: Detector effect on the scores for WBF Higgs production, for fixed functional forms derived at parton level.

| $f_{W\widetilde{W}} = 0$ Eq.(39) | parton level | detector | pull |
|---|---|---|---|
| $a$ | 1.086(11) | 0.9264(20) | 14.5 |
| $b$ | 0.10241(19) | 0.08387(35) | 97.6 |
| $c$ | 0.24165(84) | 0.3542(20) | 134.0 |
| $d$ | 0.00662(32) | 0.00911(67) | 7.75 |
| MSE | $8.50 \cdot 10^{-3}$ | $1.51 \cdot 10^{-2}$ | |

| $f_{W\widetilde{W}} = 1$ Eq.(41) | parton level | detector | pull |
|---|---|---|---|
| $a'$ | 0.7490(14) | 0.8792(31) | 93.0 |
| $b'$ | 0.37800(94) | 0.4160(19) | 40.4 |
| $c'$ | 4.218(18) | 3.526(31) | 38.4 |
| $d'$ | 4.598(18) | 4.759(32) | 8.9 |
| $e'$ | 1.1271(26) | 1.0950(48) | 1.2 |
| $f'$ | -0.2638(49) | -0.2325(68) | 6.4 |
| $g'$ | 0.2063(19) | 0.2057(34) | 0.3 |
| MSE | $3.89 \cdot 10^{-2}$ | $4.15 \cdot 10^{-2}$ | |

effects on the actual output, it is most useful to compare expressions after the optimization fit of the PYSR output. In the left part of Tab. 10 we compare the two sets of coefficients. The main aspects from the previous discussions still hold, $d \ll 1$ ensures $t \propto \sin\Delta\phi$ also after detector effects, and $b/a \ll 1$ limits the impact of the rapidity observable. The shift in the best values for the four parameters is statistically significant, but in practice most likely negligible.

For the more complex case of $f_{W\widetilde{W}} = 1$, where we do not have a closed form for the theory description, the detector effects on the PYSR convergence are more severe. However, as long as the detector effects do not change the final state particles we can again fit the parton-level formula of Eq.(41) to the detector-level score given by MADMINER. In the right part of Tab. 10 we confirm the picture for $f_{W\widetilde{W}}$. While the individual coefficients change in a statistically significant manner, the general picture is unchanged. In practice, these results imply that once we have an established and understood PYSR result for scores at the parton level, we can relatively easily re-optimize them for the detector level.

## 4.4 Exclusion limits

Throughout our derivation and discussion of symbolic regression approximating the score as a function of phase space we always use the MSE defined in Eq.(9) as our figure of merit. This value indeed measures how well the analytic formulas approximate the numerically defined score distribution, but it is not clear how it is related to the performance of this score formula in an actual analysis. The reason is that the relevant phase space regions for an analysis are not necessarily the phase space regions contributing to the MSE. Quite the opposite, we generally expect tails of kinematic distributions to dominate SMEFT analyses, while not giving large contributions to the global MSE value.

To benchmark the performance of different (optimal) observables we compute the log-likelihood distribution and extract the $p$-value for an assumed $f_{W\widetilde{W}} = 0$ including detector effects and for an integrated LHC luminosity of 139 fb$^{-1}$. We start with the analytic functions

$$a_1 p_{T,1} p_{T,2}, \qquad a_2 \sin\Delta\phi, \qquad a_3 p_{T,1} p_{T,2} \sin\Delta\phi, \qquad (42)$$

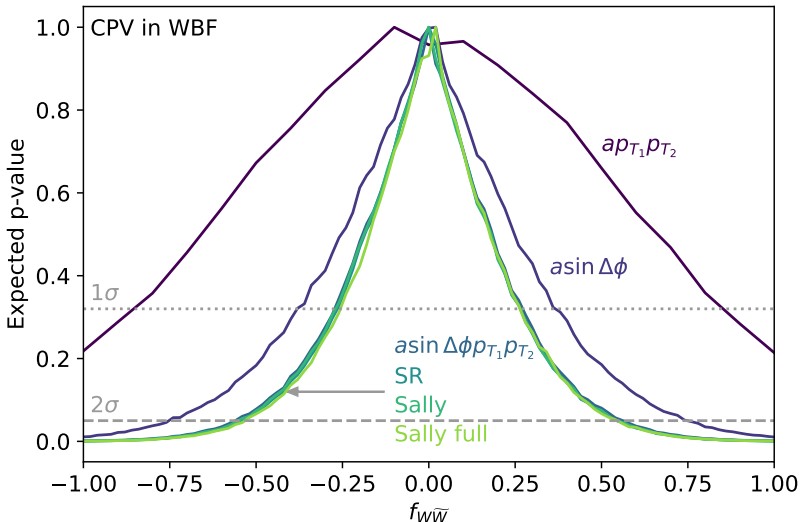

Figure 12: Projected exclusion limits assuming $f_{W\widetilde{W}} = 0$ for different (optimal) observables. The SALLY network uses $p_{T_1}$, $p_{T_2}$, $\Delta\phi$ and $\Delta\eta$, Sally full uses 18 kinematic variables.

with $a_1 = -8.32(89)\cdot 10^{-7}$, $a_2 = -0.37370(94)$, and $a_3 = -5.5386(49)\cdot 10^{-5}$ and compare the results to the reach of the complete SR expression of Eq.(39). Finally, we compare these results to the SALLY method using the four PySR observables in Eq.(38), and using the full set of 18 observables. The exclusion limits are shown in Fig. 12 and in Tab. 11. First, we confirm that for all score approximations the likelihood follows a Gaussian shape. Second, we find that beyond the minimal reasonable form $a p_{T1} p_{T2} \sin\Delta\phi$ there is only very little improvement in the expected LHC reach.

The plateau we observe in the expected exclusion limits indicates that an improved description of the score over all of phase space does not automatically result in an improved reach. Events with high scores in kinematic tails are rare and therefore contribute little to the global MSE value, but they are crucial for the actual measurement. In contrast, events with low scores in the kinematic bulk dominate the MSE, but hardly affect our specific SMEFT measurement of $f_{W\widetilde{W}}$. This means that the MSE is an orthogonal and typically more sensitive figure of merit for our symbolic regression task. To understand the different behaviors of the expected limit and the MSE we divide phase space into different score regions and compute the score for all events, events with intermediate score values $|t(f_{W\widetilde{W}})| = 0.1 \dots 0.5$, and event

Table 11: MSE and exclusion limits for different approximations of the score or candidate optimal observable. The different scenarios correspond to Fig. 12.

| (optimal) observable | MSE | | | | reach | |
|---|---|---|---|---|---|---|
| | all | $|t(f_{W\widetilde{W}})| = 0.1 \dots 0.5$ | $|t(f_{W\widetilde{W}})| > 0.5$ | weighted | $1\,\sigma$ | $2\,\sigma$ |
| $a p_{T1} p_{T2}$ | 0.1576 | 0.0645 | 1.144 | 0.298 | [-0.86,0.86] | — |
| $a\Delta\sin\phi$ | 0.0885 | 0.0163 | 0.680 | 0.223 | [-0.38,0.36] | [-0.76,0.74] |
| $a\Delta\sin\phi\, p_{T1} p_{T2}$ | 0.0217 | 0.0076 | 0.163 | 0.056 | [-0.28,0.28] | [-0.56,0.56] |
| SR Eq.(39) | 0.0145 | 0.0059 | 0.103 | 0.031 | [-0.26,0.26] | [-0.54,0.54] |
| SALLY | 0.0129 | 0.0051 | 0.092 | 0.030 | [-0.26,0.26] | [-0.56,0.54] |
| SALLY full | 0.0048 | 0.0031 | 0.026 | 0.014 | [-0.26,0.26] | [-0.54,0.54] |

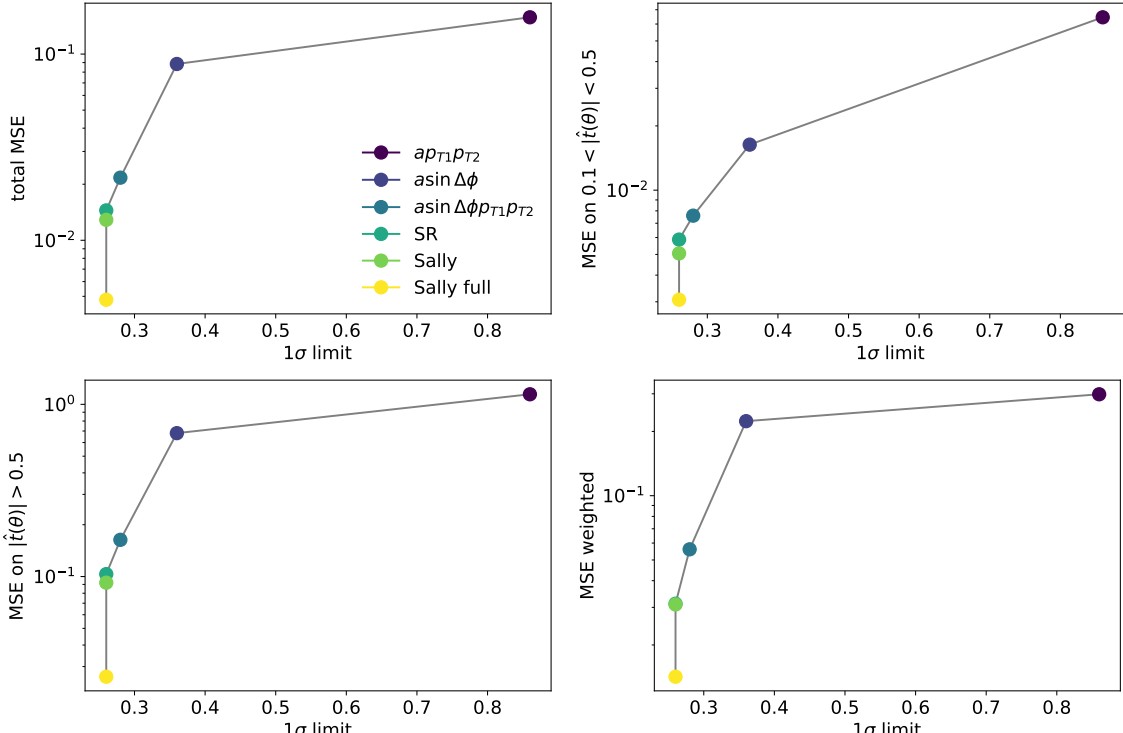

Figure 13: Scaling of the expected exclusion limits with the MSE for the four MSE evaluations defined in Tab. 11.

with large score $|t(f_{W\widetilde{W}})| > 1$ in Tab. 11. We also compute a score-weighted MSE as

$$\text{MSE}_{\text{weighted}} = \frac{1}{n}\sum_{i=1}^{n} g_i(x)(g_i(x) - t_i(x, z|\theta))^2 \ . \tag{43}$$

The correlation between the MSE and the different scores are illustrated in Fig. 13. All MSE definitions share the common feature that a strong MSE–score correlation for the simple approximate formulas becomes flat when we reach the simplified formula $t \propto p_{T1} p_{T2} \sin\Delta\phi$ and the closed formula from PySR. While we observe a slight improvement in all MSE definitions by going to the full, numerically defined SALLY network, this improvement appears to have no impact on a possible analysis.

Nevertheless, Fig. 13 illustrates a way to use our new approach in an actual LHC analysis like the one of Ref. [31]. Right now we have no option in between using the approximate optimal observable given in Eq.(5) and the computing-intensive SALLY framework. An analytic observable which matches the SALLY results also at higher statistical precision, for this reference value $\theta_0$ or others an for this channel or others, it would not only simplify the analysis setup, it would also render such an analysis much more transparent in the sense of interpretable numerics and machine learning. While interpretability might not seem very relevant for limit setting, this aspect will become crucial once a measurements points to physics beyond the Standard Model.

## 5 Outlook

Modern machine learning opens extremely promising new avenues in experimental and theoretical particle physics, but has the disadvantage of only providing numerical functions. Tra-

ditionally, theoretical and experimental particle physics work with approximate formulas provided by perturbation series in quantum field theory. Symbolic regression combines the benefits of machine learning and analytic formulas by learning complex functions from low-level or high-dimensional data and expressing them analytically.

In this first application of symbolic regression to LHC simulations (see also Ref. [53]) we use a genetic algorithm implemented in PYSR [28] to extract optimal observables or the score as an analytic function of phase space observables. The input to the PYSR training is the matrix element used for standard LHC simulations. Our theory parameters of interest are individual SMEFT Wilson coefficients. First, we study the coefficient $f_B$ in a toy setup of $ZH$ production and extract a simple polynomial for the score around the SM value $f_B = 0$. For larger values of $f_B = 10/\text{TeV}^2$ the task becomes more challenging because of saturation effects, so PYSR resorts to rational functions. For the $ZH$ production example we illustrate how the score is computed from the joint score, including multiple topologies and unobservable parameters like the flavor of the incoming quarks.

For the theoretically more interesting case of $CP$-violation through the Wilson coefficient $f_{W\widetilde{W}}$ we compute the optimal observable or score for WBF Higgs production. For small Wilson coefficients our PYSR-based DEEPDIETER tool finds a compact formula for the optimal observable, including the sine-dependence on the azimuthal angle between the tagging jets and a momentum-dependent pre-factor, $p_{T,1}p_{T,2}\sin(\Delta\phi)$. To the best of our knowledge, this is the first LHC-physics formula derived using modern machine learning.[1] Again, the regression task becomes significantly more complicated for large Wilson coefficients. For the WBF case we show how it is possible to include detector effects. Finally, we estimate the LHC reach for a a range of different PYSR formulas and for the neural networks provided by MADMINER and find that simple PYSR formulas can be used in experiment without any loss in performance.

While not all neural networks used at the LHC can and should be replaced by learned formulas, in many instances such formulas will help us understand numerical results and relate them to perturbative theory predictions. Here, symbolic regression as part of our machine learning strategy will strengthen the defining link between fundamental theory and complex experimental analyses in particle physics.

## Acknowledgments

We are very grateful to Miles Cranmer for help on PYSR, Kyle Cranmer for his insights, and Markus Schumacher for enlightening and fun discussion on optimal observables and how to use them in Higgs physics.

**Funding information** The research of AB and TP is supported by the Deutsche Forschungsgemeinschaft (DFG, German Research Foundation) under grant 396021762 – TRR 257 *Particle Physics Phenomenology after the Higgs Discovery*.

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
