# Peer review of "Back to the Formula -- LHC Edition"

_SciPost Physics, doi:SciPost Phys. 16, 037 (2024)_

## Round 2 · Referee Report · Jack Araz (Referee 1) · 2022-2-15

Report

In this study, the authors present a Machine Learning based symbolic regression method in order to extract analytic functions from LHC event shapes. The study is interesting and worthy of publication, but I would like to raise a few issues regarding the text.

The language of the study is highly casual and lacks various references or explanations of the concepts out of the knowledge of a general particle physics audience.

  • For instance, just before equation 2 "Taylor expansion" has been used as a verb, i.e. "taylor the log likelihood".

  • Neyman-Pearson lemma and Cramer-Rao bound requires citation. Ref [2] refers to a physics analysis paper rather than the original reference. Additionally, these methods are quite unfamiliar to the article's audience; the paper can benefit from a couple of lines of explanation.

  • Which Delphes detector card is used has never been stated in the study.

  • Parsimony has not been defined. It looks like a scaling constant, but it's unclear.

  • Simulated annealing method has not been adequately cited.

  • There are multiple references to a "score" measure with various score definitions in the paper. It would be clearer if authors could refer to function or method name or reference to an equation instead of repeatedly using the word "score".

  • EWDim6 model has not been cited.

  • Page 14 first paragraph reads, "...Eq 12 or Eq 28 do not accept enough more complex functions..." I believe there is a typo in enough/more where only one of them should be there.

  • Delphes module internally uses FastJet. Although the antikt algorithm has been cited, FastJet has never been mentioned in the paper.

I also have one additional question: representation is one of the biggest problems of ML methods where even with the best optimization algorithms and statistically valid data, an NN may not be able to represent the given model. Since in regular classification or regression type of analyses, the change in the results can be measured by training the network multiple times and taking the mean of the results, which allows a better approximation of the final result. However, since this study aims to describe the data with an analytic function, for each training, the functional form should change. A similar averaging or ensembling method would be sufficient for the cases that this difference can be absorbed into the constants. But if each execution leads to a slightly different functional form, I can not imagine how to write the analytic function to be more representative of the model. I was wondering if the authors have faced such differences in the results; if they did, do they have a method to combine these results somehow?

  • validity: -
  • significance: -
  • originality: -
  • clarity: -
  • formatting: -
  • grammar: -

Author:  Tilman Plehn  on 2022-04-21  [id 2403]

(in reply to Report 1 by Jack Araz on 2022-02-15)

For instance, just before equation 2 "Taylor expansion" has been
used as a verb, i.e. "taylor the log likelihood".
-> We believe that this phrase is perfectly understandable. We also went
through the paper to make sure we are not too sloppy. We are happy to
change things if something comes up during the editorial steps.

Neyman-Pearson lemma and Cramer-Rao bound requires citation. Ref [2]
refers to a physics analysis paper rather than the original
reference. Additionally, these methods are quite unfamiliar to the
article's audience; the paper can benefit from a couple of lines of
explanation.
-> We cited the original references and also some physics-targeted
discussions, in case readers need more details. Adding a very brief
explanation, we believe, would not help either the expert nor the
non-expert readers, and a proper explanation would be tedious on
the readers.

Which Delphes detector card is used has never been stated in the study.
-> We use the default CMS card, now mentioned in the text.

Parsimony has not been defined. It looks like a scaling constant,
but it's unclear.
-> We made it explicit that it is defined through exactly this
formula.

Simulated annealing method has not been adequately cited.
-> Now citing the original paper.

There are multiple references to a "score" measure with various
score definitions in the paper. It would be clearer if authors could
refer to function or method name or reference to an equation instead
of repeatedly using the word "score".
-> We are aware of this problem, but `score' is the technical term
used in the statistics community for what we call optimal
observables. We clarified this more after Eq.(2), with an original
reference to Fisher.

EWDim6 model has not been cited.
-> We added the reference.

Page 14 first paragraph reads, "...Eq 12 or Eq 28 do not accept
enough more complex functions..." I believe there is a typo in
enough/more where only one of them should be there.
-> We might need some language help from the journal concerning
commas. We actually mean enough of more complex functions. How
would we write this better?

Delphes module internally uses FastJet. Although the antikt
algorithm has been cited, FastJet has never been mentioned in the
paper.
-> We added a reference.

I also have one additional question: representation is one of the
biggest problems of ML methods where even with the best optimization
algorithms and statistically valid data, an NN may not be able to
represent the given model. Since in regular classification or
regression type of analyses, the change in the results can be measured
by training the network multiple times and taking the mean of the
results, which allows a better approximation of the final
result. However, since this study aims to describe the data with an
analytic function, for each training, the functional form should
change. A similar averaging or ensembling method would be sufficient
for the cases that this difference can be absorbed into the
constants. But if each execution leads to a slightly different
functional form, I can not imagine how to write the analytic function
to be more representative of the model. I was wondering if the authors
have faced such differences in the results; if they did, do they have
a method to combine these results somehow?
-> That is an interesting question, but going beyond what we can do
here. As written in the paper, we targeted this problem in two
ways. First of all, we always report the hall of fame with the
different possible formulas. Secondly, we added an actual parameter
fit to the hall-of-fame formulas and quote the results with error
bars. Given that we are not actually employing a neural network, we
are not sure how ensembling would help in addition to these two
modifications of the standard SR approach.

Attachment:

SR_v2.pdf

---

## Round 2 · Referee Report · Anonymous (Referee 2) · 2022-2-22

Report

This paper uses symbolic regression to identify ML-informed observables for new physics searches. I think the paper is detailed, well-written and deserves publication after a few small questions are addressed. I do commend the authors on their choice of network naming.

I do not quite see how this approach will stand its ground when it comes to scalability and portability. The processes that the authors choose to investigate in this proof-of-principle analysis are well-motivated, and I do understand that something comprehensive seems to be in the making. Nonetheless, these processes are so simple at leading order in terms of involved mass scales and kinematics that it is hard to see the approach's relevance going forward. While it is good to see that known results are reproduced, a lot of it seems to rely on an educated choice of input parameters? Also, how much is this approach actually limited by ab initio choices? Given that the information extracted by the network is essentially the correlation of the matrix element, what is the selling point of this approach compared to e.g. the matrix element method?

Higher order corrections will introduce non-polynomial uncertainties and dependencies, is this a bottleneck for this approach?
  • validity: -
  • significance: -
  • originality: -
  • clarity: -
  • formatting: -
  • grammar: -

Author:  Tilman Plehn  on 2022-07-24  [id 2681]

(in reply to Report 2 on 2022-02-22)
Category:
answer to question

-> First of all, I have to apologize for having forgotten to answer
this referee report.

This paper uses symbolic regression to identify ML-informed
observables for new physics searches. I think the paper is detailed,
well-written and deserves publication after a few small questions are
addressed. I do commend the authors on their choice of network naming.

-> Thank you :)

I do not quite see how this approach will stand its ground when it
comes to scalability and portability. The processes that the authors
choose to investigate in this proof-of-principle analysis are
well-motivated, and I do understand that something comprehensive seems
to be in the making. Nonetheless, these processes are so simple at
leading order in terms of involved mass scales and kinematics that it
is hard to see the approach's relevance going forward.

-> We beg to disagree with this judgement for the second WBF
process. For this process there exists an ATLAS analysis [31] using
an optimal observable extracted from the corresponding matrix
elements. The way forward is illustrated in Fig.12, where we show
that we can use improved formulas to match the performance of the
much more elaborate Sally method. We have added a brief paragraph
discussion this to the end of Sec.4.

While it is good to see that known results are reproduced, a lot of it
seems to rely on an educated choice of input parameters?

-> The educated choice is really only relevant for our didactic
approach. Because symbolic regression scales much more favorably
than for instance a polynomial fit we can always run with a
relatively large set of observables. We have added a few sentences
to this effect after Eq.(38).

Also, how much is this approach actually limited by ab initio choices?
Given that the information extracted by the network is essentially the
correlation of the matrix element, what is the selling point of this
approach compared to e.g. the matrix element method?

-> If the matrix element method works, it is guaranteed to extract all
available information from the data, while our method will always
provide an interpretable approximation. The great advantage of any
optimal observable method is, however, that it is experimentally
much easier and much less CPU-intensive. The question about the
full matrix element method would have to be asked for Sally, and it
is not clear to us which of the two methods will work better and be
more stable in practice. A simple optimal observable, especially in
terms of a formula, is an infinitely simpler analysis.

Higher order corrections will introduce non-polynomial uncertainties
and dependencies, is this a bottleneck for this approach?

-> Extra jets will be a challenge to the choice of observables, and it
would be very interesting to run such a study to find out what the
actually relevant observables for the hard process would be. We
added a few sentences along those lines to Sec.4.3.

Attachment:

SR_v3.pdf

---

## Round 3 · List of Changes

All comments by both referees now incorporated, see individual referee responses.

---

## Editorial Decision

published